# Modulatory effects of rutin and vitamin A on hyperglycemia induced glycation, oxidative stress and inflammation in high-fat-fructose diet animal model

Aqsa Iqbal[1], Sairah Hafeez Kamran[1]*, Farhan Siddique[2], Saiqa Ishtiaq[3¤], Misbah Hameed[4], Mobina Manzoor[4]

1 Faculty of Pharmaceutical and Allied Health Sciences, Department of Pharmacology, Institute of Pharmacy, Lahore College for Women University, Lahore, Punjab, Pakistan, 2 Faculty of Pharmacy, Department of Pharmaceutical Chemistry, Bahauddin Zakariya University, Multan, Punjab, Pakistan, 3 Punjab University College of Pharmacy, University of the Punjab, Allama Iqbal Campus, Lahore, Punjab, Pakistan, 4 Faculty of Pharmaceutical and Allied Health Sciences, Department of Pharmaceutics, Institute of Pharmacy, Lahore College for Women University, Lahore, Punjab, Pakistan

¤ Current address: Center for the Study of Human Health, Emory College of Art and Science, Emory University, Atlanta, GA, United States of America

* sairah.hafeez@lcwu.edu.pk

**Data Availability Statement:** All relevant data are within the paper and its Supporting Information files.

## Abstract

In the current study we investigated the impact of combination of rutin and vitamin A on glycated products, the glyoxalase system, oxidative markers, and inflammation in animals fed a high-fat high-fructose (HFFD) diet. Thirty rats were randomly divided into six groups (n = 5). The treatments, metformin (120 mg/kg), rutin (100 mg/kg), vitamin A (43 IU/kg), and a combination of rutin (100 mg/kg) and vitamin A (43 IU/kg) were given to relevant groups of rats along with high-fructose high-fat diet for 42 days. HbA1c, D-lactate, Glyoxylase-1, Hexokinase 2, malondialdehyde (MDA), glutathione peroxidase (GPx), catalase (CAT), nuclear transcription factor-B (NF-κB), interleukin-6 (IL-6), interleukin-8 (IL-8) and histological examinations were performed after 42 days. The docking simulations were conducted using Auto Dock package. The combined effects of rutin and vitamin A in treated rats significantly (p < 0.001) reduced HbA1c, hexokinase 2, and D-lactate levels while preventing cellular damage. The combination dramatically (p < 0.001) decreased MDA, CAT, and GPx in treated rats and decreased the expression of inflammatory cytokines such as IL-6 and IL-8, as well as the transcription factor NF-κB. The molecular docking investigations revealed that rutin had a strong affinity for several important biomolecules, including as NF-κB, Catalase, MDA, IL-6, hexokinase 2, and GPx. The results propose beneficial impact of rutin and vitamin A as a convincing treatment strategy to treat AGE-related disorders, such as diabetes, autism, alzheimer's, atherosclerosis.

**Funding:** The author(s) received no specific funding for this work.

**Competing interests:** The authors have declared that no competing interests exist.

**Abbreviations:** AGE, Advanced glycation end products; BSA, Bovine Serum Albumin; CAT & 1DGB, Catalase; ELISA, Enzyme-linked immunosorbent assay; HFFD, High Fat Fructose Diet; Glo1 & 7WT1, Glyoxalase I; Glo2, Glyoxalase II; GPx & 2P31, Glutathione peroxidase; GSH, Glutathione; IL-6 & 2L3Y, Interleukin-6; IL-8 & 5D14, Interleukin-8; MDA & 1HZ2, Malondialdehyde; MG, Methylglyoxal; NF-κB & 1A3Q, Nuclear factor kappa B; NADH, Nicotinamide adenine dinucleotide hydrogen; qRT-PCR, Quantitative reverse transcriptase polymerase chain reaction; ROS, Reactive Oxygen Species; RAGE, Receptor for advanced glycation end products; SOD, Superoxide dismutase; SEM, Standard error of mean; TNF- α, Tumor necrosis factor- alpha; 2NZT, Hexokinase 2.

# Introduction

Hyperglycemia, oxidative stress, inflammation, and glycated products are interconnected factors that play significant roles in the development and progression of diabetes and its associated complications. In some situations, such as those involving stress, illness, or the consumption of a lot of carbohydrates and fat, hyperglycemia can also happen in people without diabetes [1,2]. Hyperglycemia and hyperlipidemia can contribute to oxidative stress leading to an imbalance between the production of reactive oxygen species (ROS) and the body's ability to neutralize or detoxify them. Excessive generation of ROS leads to a rise in glycated products, which are highly reactive molecules that can damage cells, proteins, lipids, and DNA. This damage contributes to tissue injury and dysfunction [3]. Three main pathways lead to oxidative stress and elevated inflammatory status: first is the binding of advanced glycation end products (AGEs) to protein or cell surfaces; second is the production of ROS and the accumulation of AGEs; and third is the interaction of AGE with AGE receptors (RAGE), which sets off a chain reaction of inflammatory mediators. Formation of AGEs over an extended period induces the formation of the pro-inflammatory effector molecules, as well as higher levels of TNF-α, IL-8, IL-6, IL-1β [4]. Early glycation products such as methylglyoxal (MG), glyoxal (GO), and 3-deoxyglucosone (3-DG) can be converted into more stable AGEs, which may be irreversibly cross-linked with proteins or DNA and may subsequently alter the organ function and structure [5]. Stable AGEs accumulate in tissues over time, leading to damage to cells and tissues, and are associated with a wide range of chronic diseases, including diabetes, cardiovascular disease, Alzheimer's, and some cancers [6]. The glyoxalase system plays a crucial role in the metabolism of stable AGEs. Glyoxalase 1 (Glo1) and Glyoxalase 2 (Glo2) make up the glyoxalase system. Methylglyoxal is converted by the enzyme Glo1 into S-D-lactoylglutathione with the help of the reduced form of glutathione (GSH) as a cofactor. This detoxification procedure aids in preventing methylglyoxal buildup and the damaging impact it has on DNA, lipids, and cellular proteins. S-D-lactoylglutathione is hydrolyzed by Glo2 to generate D-lactate, which restores the GSH that was depleted during Glo1 catalysis. Glo1 is especially important in cells and tissues exposed to high levels of oxidative stress, such as those with increased glucose metabolism, as seen in diabetes [7,8]. Research has shown that Glo1 expression and activity can vary among individuals and can influence susceptibility to certain diseases, including diabetes and its complications. Some studies have explored the potential therapeutic implications of targeting the glyoxalase system to mitigate oxidative stress and the harmful effects of methylglyoxal [9]. Around the world, many diets (including soft drinks and many pre-packaged meals) include sugars like glucose and fructose, which can make up to 60% of total sugar content and also comprises high-fat content [10]. Long-term consumption of high sugar, high-fat diet can result in AGE products formation, insulin resistance, diabetes, adipose tissue malfunction, obesity, and other metabolic diseases [11,12].

Flavonoids are polyphenolic compounds present in plant-based foods and they are known for their antioxidant and anti-inflammatory properties and have been linked to various health benefits. There are many different types of flavonoids, including rutin, quercetin, kaempferol, catechins, and anthocyanins. Rutin (quercetin-3-O-rutinoside) is widely present in various plants part like leaves, flowers, roots, and skin of apples, buckwheat, tea, and passionflower [13]. It is commonly used as a dietary supplement to support cardiovascular health, strengthen blood vessels, and improve circulation. It is an essential part of food's nutrition and is referred to as, rutoside, and sophorin. The pharmacological potential of rutin has been diversely researched including antioxidant, anti-obesity, anti-diabetic, cytoprotective, vasoprotective, anticarcinogenic, antifungal, antibacterial, antiviral, neuroprotective, nephroprotective, hepatoprotective, immunomodulatory and cardioprotective activities. In the gut, enzymatic

hydrolysis of rutin by α-glucosidase produces quercetin and rutinose [14]. Rutin has been studied for its ability to prevent AGE formation and *in-vitro* studies have shown that rutin prevents AGEs formation in rat muscle and kidney. Rutin prevented the growth of AGE on proteins in the eye lens and dietary G-rutin could lower the amount of AGE proteins in serum and kidney in diabetic rats. Rutin ameliorated lipid peroxidation, decreased levels of ROS and oxidized glutathione, increased levels of glutathione peroxide and reduced glutathione, and decreased superoxide dismutase activity in the renal tissue, according to several other investigations. It also decreased the collagen and hydroxyproline levels and increased matrix metalloproteinase activity in the kidney of diabetic rats [15]. Vitamins are essential for good health and play an important role in various physiological functions. A fat-soluble vitamin, vitamin A is important for immunity, brain cell development, reproduction, and especially eye health. It also plays a significant role in the conditions of obesity, dyslipidemia, diabetes, and kidney damage [16]. Vitamin A is crucial for preserving cellular redox equilibrium. According to several experts, vitamin A lowers the synthesis of NF-κB [17]. Retinol, a vitamin A component, has been demonstrated to upregulate RAGE by free radical-dependent activation of p38 and Akt. It also plays a crucial function in shielding biomolecules from oxidative damage induced by ROS [18]. The evaluation of the role of rutin and vitamin A on the hexokinase and glyoxalase system has not yet been conducted. The goal of the current study was to evaluate the effects of rutin and vitamin A alone and in combination on the biochemical markers by examining the glyoxalase system, inflammatory and oxidative stress indicators in HFFD fed animal model.

## Materials and methods

### Materials

Bovine serum albumin (Sigma Aldrich, USA), Fructose (Sigma Aldrich, USA), Sodium Azide (Sigma Aldrich, USA), Rutin (Sigma Aldrich, USA), Vitamin A (GNC, USA), Metformin (Martin Dow Pharmaceuticals (PAK) LTD), and Nitro-blue tetrazolium (NBT) (Sigma Aldrich, USA) were among the chemicals bought from different suppliers.

### *In-silico* molecular docking studies

Two drug candidates Rutin and vitamin-A were retrieved from the PubChem database [19]. For molecular docking, MGLtools, the Autodock4 and Autogrid4 [20] binary files, the Discovery Studio Visualizer from BIOVIA, ChemDraw ultra (*ChemDraw Ultra 12.0 0*), and ChemDraw 3D pro (*Chem 3D Pro 12.0*) were utilized. The target proteins were obtained from RCSB and then processed so that the autodock suite could be run on it. In BIOVIA's discovery studio visualizer, we eliminated all heteroatoms as well as co-crystal ligand and solvent molecules from the protein molecule. After obtaining a clean protein structure, autodock tools were used to optimize it for docking by giving each atom the appropriate polar hydrogen and Kollman charges and saving the file as a pdbqt [21]. Using the compounds' IUPAC designations, the structures were drawn in Chemdraw ultra, and then the energy minimization was performed in chem 3D pro. SDF files with the compound structures have been stored. After that, the openbabel GUI program was used to transform these structures into a pdbqt file, which is a format that Autodock can read. The investigated target proteins names (PDB ID) were NF-κB (1A3Q), catalase (1DGB), malondialdehyde (1HZ2), glutathione peroxidase (2P31), IL-6 (2L3Y), IL-8 (5D14), glyoxalase-1 (7WT1), hexokinase 2 (2NZT). Autodock4 was then used to do structure-based virtual screening. For creating the grid parameter file, the box dimensions were set. Docking parameter files were generated using a combination of a custom force field called Autodock4Zn and a Lamarckian genetic algorithm (LGA). To guarantee the highest

level of accuracy, we used a population size of 300 and several postures of 50 [22]. Following preparation, the ligand library was docked independently into the active site of each protein [20]. For this purpose, we employed the ligand-protein interaction analyzer BIOVIA discovery studio visualizer. Protein pdbqt and autodock output files were imported into BIOVIA's Discovery Studio to create all the 2D and 3D conformations. It was determined which contacts between the ligand and active pocket were bonding and which were not. The RMSD value and re-docking of the co-crystal ligand into the active pocket of the protein were used to verify the docking technique. Docking and experimental ligand RMSD values less than 2.0 were required for acceptance of poses [23].

### *In-vitro* fructosamine assay

The fructosamine concentration was determined based on a previously reported method [24] with few modifications. The glycated bovine serum albumin (BSA) was prepared by incubating 10 mg/ml of BSA with 500mM fructose in phosphate buffer (0.1M at pH 7.4) also containing 0.02% sodium azide ($NaN_3$) to prevent bacterial contamination at 37˚C for 28 days in the presence or absence of rutin (12.5μM, 50μM, 50μM and 100μM), vitamin A (100μg/ml, 300μg/ml, 600μg/ml and 1200μg/ml), combination of rutin and vitamin A and metformin (1 mM). A solvent of 5% dimethyl sulfoxide (DMSO) was employed for the extraction of rutin. The samples were stored in sealed containers and maintained at a temperature of 37˚C in a light-free environment. The positive control in this study consisted of BSA + fructose, which was likewise maintained under comparable conditions [25].

The *in-vitro* fructosamine assay was performed by adding 20 μL glycated BSA sample to 180 μL of 0.5mM nitro blue tetrazolium (NBT) solution prepared in sodium carbonate buffer (0.1M, pH 10.35) and kept at room temperature for 30 min. The absorbance was measured with a microplate reader at 530 nm wavelength and percentage inhibition was calculated using formula.

$$\text{Inhibitory activity (\%)} = [(A_0 - A_1)/A_0] \times 100$$

Where $A_0$ is the absorbance value of positive control and $A_1$ is the absorbance of glycated sample.

### *In-vivo* studies

**Development of animal model and induction of hyperglycemia.** At the animal house of Lahore College for Women University, thirty male albino rats (180-220g) were purchased from the University of Veterinary and Animal Sciences (UVAS, LHR) and acclimatized for one week under standard conditions (temperature of 20–25˚C, humidity of 50–55%, and a 12-h light-dark cycle). Daily administration of HFFD (30% fat (5% sunflower oil+ 10% cottonseed oil+ 15% saturated fatty acids), 16% proteins, calcium (5%), vitamins mixture (1%), casein (11%), amino acids mixture (22%) and 35% fructose solution (10 mL/kg P.O) was used to induce hyperglycemia. For 42 days, 10% fructose in drinking water was also given daily [26,27]. The test substances and the HFFD were given at the same time. Metformin (120 mg/kg) [28] and rutin (100 mg/kg) doses were chosen based on earlier research [29]. By using a conversion factor, the vitamin A dose was determined in accordance with the amount for an adult human [30].

**Ethical approval.** The experimental research protocol was approved by the Research Ethics Institutional Review Board, **LCWU (ORIC/LCWU/410)**

**Study design.** Six groups of animals (n = 5) were created, and the corresponding treatments were given for 42 days. Group 1 (the negative control group) got a regular diet and

water. HFFD was administered to Group 2 (the disease control group). Group 3 received HFFD plus Metformin (120 mg/kg) as standard treatment. Group 4 was administered HFFD and Rutin (100 mg/kg). Group 5 obtained HFFD, and 43 IU/kg of vitamin A. Group 6 was administered HFFD along with 100 mg/kg of rutin and 43 IU/kg of vitamin A. All the groups except for the negative control were provided with 10% fructose in drinking water for 42 days. Following the final dose, rats were anesthetized with 0.2 mL/kg intra-peritoneal injection of cocktail of xylazine and ketamine (2.5 mL of xylazine, 50 mg/mL, 10 mL of ketamine) following an overnight fast. Responses were evaluated after five minutes of anesthesia to confirm the anesthetic stage. Following the confirmation of anesthesia, rats were euthanized using cardiac puncture, and blood and tissues were subsequently collected for further analysis [31].

**Measurement of body weight and fasting glucose level.** The body weight of each rat was measured at the end of the study using digital weighing balance. Using an Accu-check glucometer, fasting blood glucose levels were tested at the 42$^{nd}$ day of the study.

**Biochemical parameters.** IchromaTM HbA1c Kit (Boditech, Korea) was used to measure glycated HbA1c in whole blood. Rat Hexokinase-2 (HK2) ELISA Kit (Bioassay Technology Laboratory, China) was used to measure the levels of hexokinase-2 in the liver tissue. The degree of glyoxalase expression in rat liver tissue was assessed using the Rat Glo1 ELISA Kit (Elabscience, USA).

**Antioxidant parameters.** Using the ELISA technique, malondialdehyde was assessed in rat tissue [32] With minor adjustments, the Moron technique was used to measure glutathione peroxidase. The tissue homogenate 100μL was treated with 5 mM GSH, 1.2 mM $H_2O_2$, 25 mM $NaN_3$, and phosphate buffer (1 M, pH 7.0) in a total volume of 2.5 mL at a temperature of 37˚C for a duration of 6 minutes. The reaction was stopped upon the addition of 2.0 mL of phosphoric acid at a concentration of 1.65%. The reaction mixture was centrifuged for 10 minutes at 3000 rpm. The supernatant obtained was mixed with 0.4 M of $Na_2HPO_4$ solution and 1 mM DTNB (5,5′-dithiobis-(2-nitrobenzoic acid) solution prepared in phosphate buffer. The solution was subjected to a 10-minute incubation period at a temperature of 37˚C, and the absorbance was quantified at a wavelength of 412nm [33]. The method of Beers and Sizer was slightly modified to measure catalase activity [34]. The catalase activity was determined by adding 1.9 mL of phosphate buffer and 1 mL of $H_2O_2$ solution to 100μL of prepared tissue homogenate. At 240 nm, the extinction was recorded at intervals of 1 and 3 minutes. The determination of catalase activity was conducted using the molar extinction coefficient, which was found to be 43.6.

**Real-time PCR assay of cytokines.** The expression levels of the pro-inflammatory cytokines IL-6, IL-8, and NF-b in liver tissue samples were determined using the qRT-PCR method. Total RNA was isolated from tissue samples utilizing a multi-type of sample DNA/RNA Extraction-Purification Kit (Sansure Biotech Inc, China). Then, additional reverse transcription into cDNA was performed using a Thermo-Fisher scientific cDNA kit (HRP013 100T) from Zokyo, China. The qRT-PCR technology was used in accordance with the recommended method. In a SLAN-96P Real-Time PCR System (Sansure Biotech Inc., China), qRT-PCR was carried out using a 2X SYBR qPCR Mixture (ZOKEYO, China). The total reaction volume of 15 μL, which included 10μL of SYBR Green mix, primers at a concentration of 0.5 mM each, and 1 μL of cDNA as a template with the appropriate primers and housekeeping genes (GAPDH) were employed as controls for comparing the relevant Cts of samples to controls. The amplification conditions were 95˚C for 30 s and 40 cycles at 95˚C for 5 s and 60˚C for 20 s [35]. The primer sequence of the gene NF-κB, IL-8 and IL-6 are presented in **Table 1**.

**Histopathological study.** Each group's liver and pancreas tissue were excised at the end of experiment, washed with saline and preserved in 10% formalin (Sigma-Aldrich, St. Louis,

**Table 1. Primer sequence used for qRT-PCR.**

| Gene | Primer | Sequence |
|---|---|---|
| NF-κB | ACGCAAAAGGACCTACGAGA | Forward |
| | ATGGTGCTGAGGGATGTTGA | Reverse |
| IL-8 | GAGAGTGATTGAGAGTGGACCAC | Forward |
| | CACAACCCTCTGCACCCAGTTT | Reverse |
| IL-6 | CCCACCAGGAACGAAAGTCA | Forward |
| | ACTGGCTGGAAGTCTCTTGC | Reverse |

MO, USA). The tissues were divided into sections and stained with hematoxylin and eosin (H&E) and viewed with a microscope (Olympus) under 40X magnification [36].

## Statistical analysis

All data were statistically analyzed, and the results were shown as Mean ± SEM (standard error of mean) (n = 5). For data analysis, the one-way ANOVA and Dunnet's post hoc test were used, and p values less than 0.05 were considered significant. When compared to the disease control group, the results of the normal control and treatment groups were marked as highly significant (***$p < 0.001$), moderately significant (**$p < 0.01$), and significant (*$p < 0.05$).

## Results

### Molecular docking study

The inhibitory potential of rutin and vitamin A against various protein targets, including NF-κB (1A3Q), Catalase (1DGB), MDA (1HZ2), IL-6 (2L3Y), hexokinase 2 (2NZT), GPx (2P31), IL-8 (5D14), and Glo1 (7WT1) were studied. Both compounds demonstrated inhibitory activity towards all the protein targets, but rutin exhibited the highest inhibitory potential against NF-κB, Catalase, MDA, IL-6, hexokinase 2, and GPx as presented in **Fig 1** and **Table 2**. This strong inhibitory potential of rutin may be attributed to the formation of multiple bonds with the amino acid moieties present at the binding sites of these proteins. Rutin displayed the highest affinity for hexokinase 2 (-9.3 kcal/mol, **Table 2**), forming six conventional H-bonds with SER70, PHE768, ASP814, H-O bond, ASP814, and SER70, Pi-donor-H bond with ARG69, three Pi-Sigma bonds with VAL459, LEU463, and LEU463, and three Pi-Alkyl bonds with ARG69, LEU163, and ARG69. It also exhibited affinity for IL-6, forming six conventional H-bonds with ASN75, LEU90, LYS91, H-O bond, GLY77, and LEU90 active pockets. It formed three Pi-alkyl bonds with ARG74, ARG74, and LEU89 as presented in **Fig 1**. The binding affinity of rutin with NF-κB was -7.7 kcal/mol, and it formed four conventional H-bonds with the amino acid moieties of NF-κB, including ARG103, GLN157, GLU92, and ASP94, one carbon-H bond, three Pi-cation, and three Pi-alkyl bonds. Among the antioxidant parameters studied, rutin displayed the highest binding affinity with catalase (-8.9 kcal/mol), forming three Pi-Alkyl bonds with ARG127 and one with VAL182. It also formed two Pi-Pi-Stacked bonds with HIS466 and one Alkyl bond with VAL126 (**S1 Table**). The flavonoid also showed H-Bond and Pi-Donor H-Bond (**Fig 1**). In contrast, the vitamin A docking studies (**Table 2**) showed a binding affinity of less than -7 kcal/mol with all molecular targets, indicating lower inhibitory potential compared to rutin. The antioxidant enzyme catalase had the highest binding affinity (-6.1 kcal/mol) with vitamin A, resulting in the formation of seven hydrophobic bonds with ALA289, PRO347, PRO347, TYR231, PHE286, PHE297, and TYR425 (**Fig 2**). Hydrophobic interactions were observed between Vitamin A and Glo-1 (-5.3), Il-8 (-5,3), and NF-κB (-4.3). Vitamin A had a binding affinity of -5.6 with MDA and established a conventional hydrogen

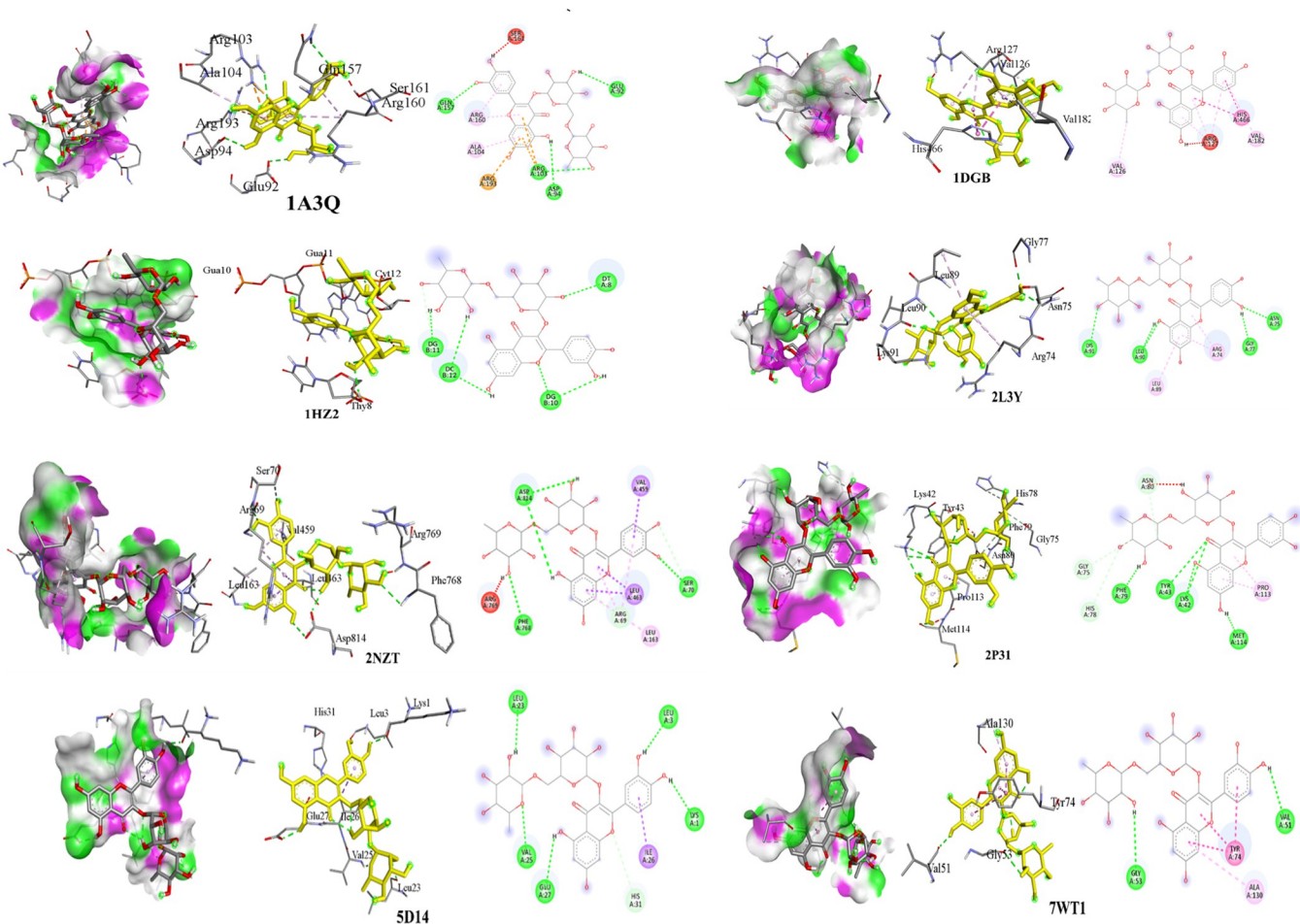

**Fig 1. Different interactions of Rutin, amino acid interactions with D-A surface, with amino acid pocket residues (left), amino acid interactions with target receptor (middle) and 2D interaction diagram (right) after molecular docking 1A3Q (NF-κB), 1DGB (Catalase), 1HZ2 (MDA), 2L3Y (IL-6), 2NZT (Hexokinase 2), 2P31 (Glutathione peroxidase), 5D14 (IL-8), 7WT1 and (Glyoxylase-1) target proteins.**

bond with the amino acid moiety DA9 at a residue distance of 2.34 Å. Vitamin A had a binding affinity of -5.0 with GPx, while it created two hydrogen bonds and one alkyl bond with the HIS78, TYR43, and PRO113 amino acid moieties respectively (**S2 Table**).

**Table 2. Binding affinity (kcal/mol) for investigated ligands rutin and vitamin A with target proteins.**

| Rutin Ligand | Binding Affinity, ΔG (kcal/mol) | Vitamin A Ligand | Binding Affinity, ΔG (kcal/mol) |
|---|---|---|---|
| 1A3Q (NF-κB) | -7.7 | 1A3Q (NF-κB) | -4.3 |
| 1HZ2 (MDA) | -7.6 | 1HZ2 (MDA) | -5.6 |
| 2NZT (hexokinase 2) | -9.3 | 2NZT (hexokinase 2) | -5.0 |
| 1DGB (Catalase) | -8.9 | 1DGB (Catalase) | -6.1 |
| 2L3Y (IL-6) | -7.9 | 2L3Y (IL-6) | -5.5 |
| 2P31 (GPx) | -8.0 | 2P31 (GPx) | -5.0 |
| 5D14 (IL-8) | -6.3 | 5D14 (IL-8) | -5.3 |
| 7WT1 (Glo1) | -6.6 | 7WT1 (Glo1) | -5.3 |

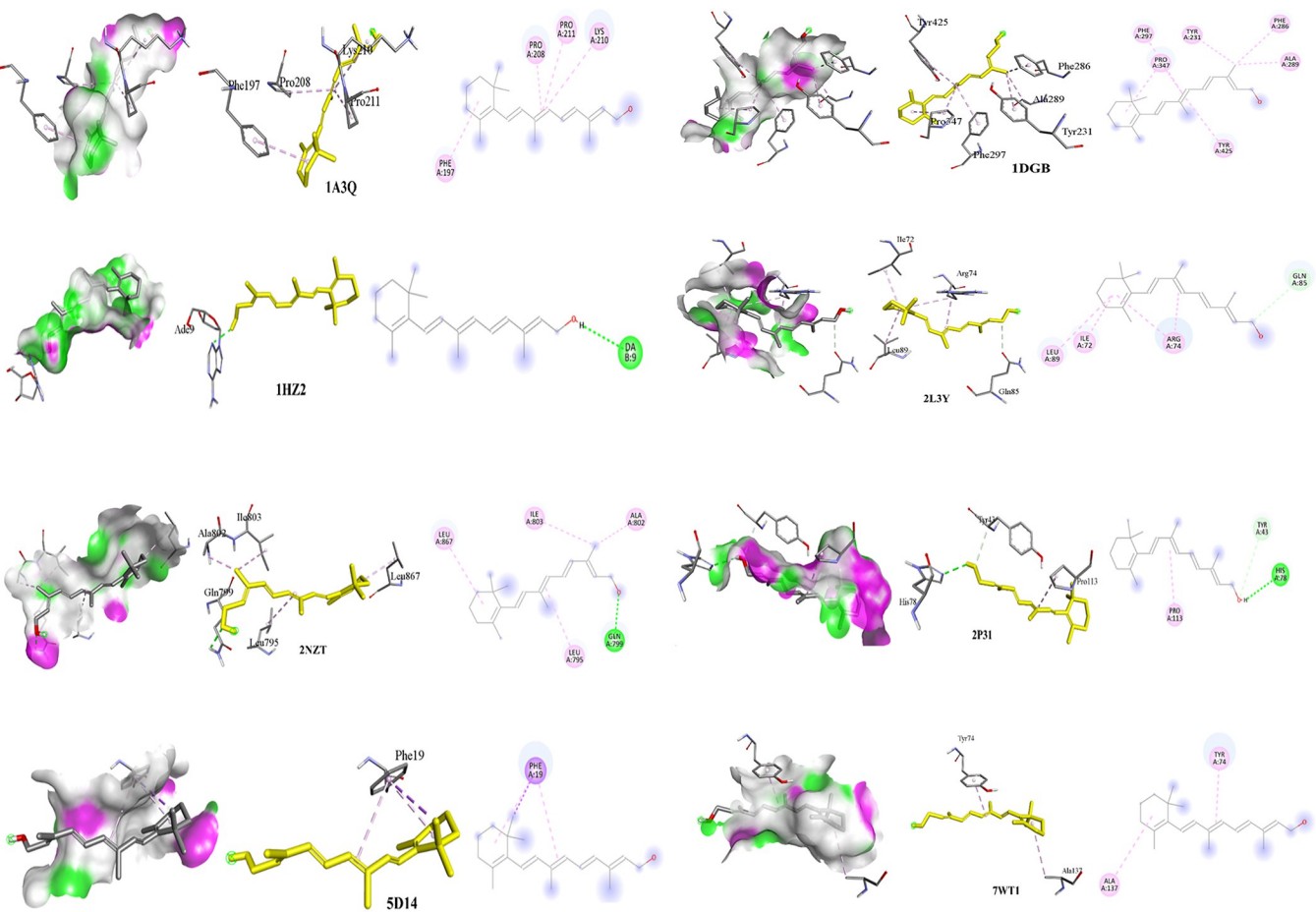

**Fig 2. Different interactions of Vitamin A, amino acid interactions with D-A surface, with amino acid pocket residues (left), amino acid interactions with target receptor (middle) and 2D interaction diagram (right) after molecular docking using 1A3Q (NF-κB), 1DGB (Catalase), 1HZ2 (MDA), 2L3Y (IL-6), 2NZT (Hexokinase 2), 2P31 (Glutathione peroxidase), 5D14 (IL-8), 7WT1 and (Glyoxylase-1) target proteins.**

### *In-vitro* fructosamine inhibition

Fructosamine inhibition exhibited by rutin (12.5–100μM), Vit A (100–1200μg/ml), their combination, metformin (1mM) and BSA were compared with BSA fructose system and presented in **Fig 3**. The combination of rutin and vitamin A exhibited remarkable inhibition of fructosamine in the assay.

### In-vivo study

**Impact of various treatments on body weights and glucose levels at the end of study.**
HFFD rats' body weight increased steadily throughout the study period, reaching its peak percentage rise on day 42 (↑19.8%) when compared to the normal control group. As shown in **Table 3**, rats receiving metformin (120 mg/kg), rutin, or a combination of rutin (100 mg/kg) and vitamin A (43 IU/kg) all showed significant ($p < 0.001$) weight loss as compared to the HFFD group. With vitamin A (↓15.61%), no notable effects were observed, and metformin has the greatest effect on weight loss. **Table 3** displays the fasting blood glucose (mg/dl) levels of all groups. On day 42, a significant difference between the normal control group and the HFFD was seen, with the difference reaching up to 346.5%, and a significant decrease in the fasting

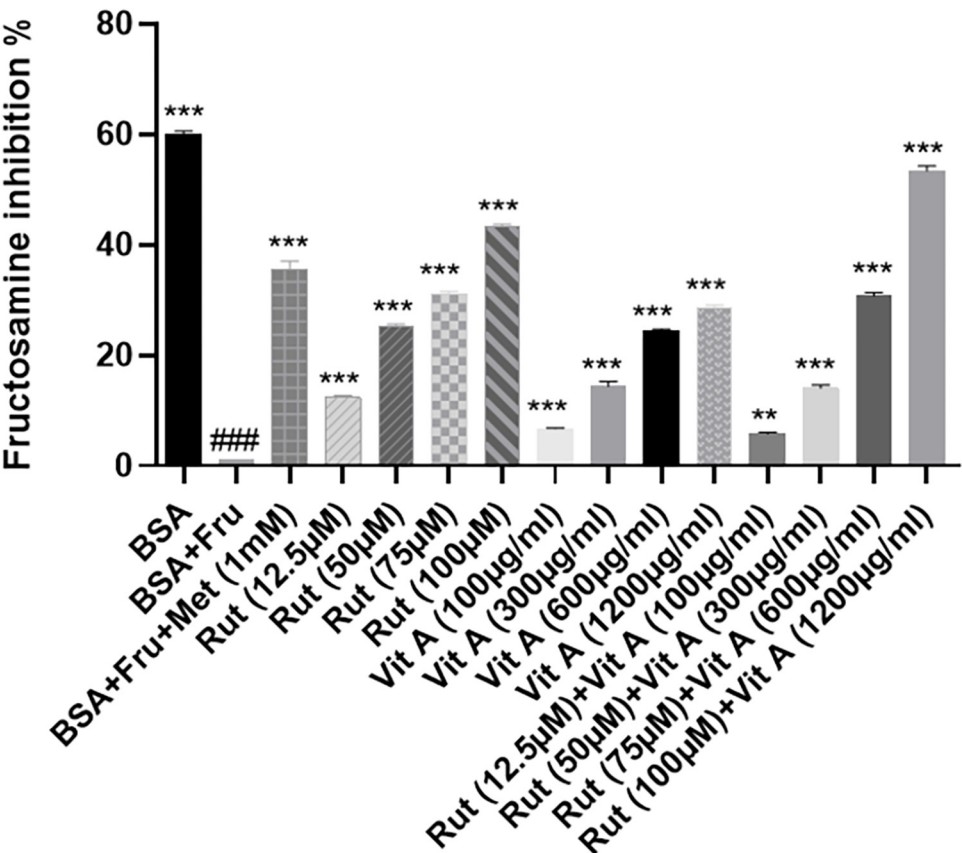

**Fig 3. Impact of BSA, Metformin (1mM), Rutin (12.5–100µM) and Vitamin A (100–1200µg/ml), on the fructosamine concentration in the BSA fructose system.** All values are expressed as mean ± SEM (n = 5) ***: p < 0.001 (when compared with BSA+Fru). ### p < 0.001 (when compared with BSA) Significant differences among treatments was observed after analysis by one way ANOVA followed by Dunnett's post-hoc test.

blood glucose levels of the various groups. When compared to the HFFD group, the blood glucose levels in the metformin (54.5%), rutin (50.5%), vitamin A (54.41%), and rutin-vitamin A (53.3%) groups all decreased.

**Effect of various treatments on advanced glycation end products, antioxidant parameters and genetic expression of inflammatory markers.** A significant increase in HbA1c level was seen in the HFFD group (7.44 ± 0.341) when compared to the normal control group (4.66 ± 0.326), and a remarkable decrease in HbA1c level of combination treatment (3.88 ± 0.171) **Fig 4A**. Hexokinase levels suppression was more evident with combination treatment (1.03 ± 0.08 ng/mL) when compared with other groups as presented in **Fig 4B**. All treatment groups showed increased Glo1 levels, which could be associated with a subsequent drop in D-lactate levels (**Fig 4C** and **4D**). All treatment groups showed an improvement in their antioxidant status, but the effects of the combination therapy were more significant. A significant decrease in GPx (1.11 ± 0.07 µmol/L) and catalase (10.2 ± 0.11 µM/g) levels of HFFD treated group was observed when compared with normal control group (3.05 ± 0.26 µmol/L & 22 ± 0.02 µM/g) and significant increase in GPx and catalase levels was observed in groups receiving metformin (2.74 ± 0.07 µmol/L & 21.9 ± 0.01 µM/g), rutin (2.72 ± 0.07 µmol/L & 13.1 ± 0.03 µM/g), vitamin A (2.77 ± 0.06 µmol/L & 15.2 ± 0.11 µM/g) and combination of rutin and vitamin A (2.84 ± 0.07 µmol/L & 21.6 ± 0.17 µM/g) (**Fig 4F** and

**Table 3. Impact of various treatments on body weights (g) and blood glucose (mg/dL) in HFFD fed rats at the end of 42 days study.**

| Groups | Weight (g) | Blood Glucose (mg/dL) |
|---|---|---|
| Normal Control | 224.6 ± 2.04* | 80.8 ± 4.07*** |
| HFFD | 269 ± 6.19# (↑19.8%) | 179 ± 6.09### (↑346.5%) |
| HFFD + Met | 180 ± 5.28*** (↓33.1%) | 81.4 ± 2.93*** (#) (↓54.5%) |
| HFFD + Rutin | 209 ± 6.39*** (↓22.3%) | 88.6 ± 1.47*** (↓50.5%) |
| HFFD + Vit A | 227 ± 20.04* (↓15.61%) | 81.6 ± 3.75*** (↓54.41%) |
| HFFD + Rutin+ Vit A | 208 ± 7.72*** (↓22.7%) | 83.2 ± 2.52*** (↓53.3%) |

Each value represents Mean ± SEM (n = 5). Two-way statistical analysis was applied in GraphPad Prism 5 and all the groups were compared with HFFD.

*: $p < 0.05$

**: $p < 0.01$

***: $p < 0.001$ (when compared with HFFD)

#: $p < 0.05$

##: $p < 0.01$

###: $p < 0.001$ (when compared with NC). Percentage reduction was calculated, ↑ presents percentage increase and ↓ shows percentage decrease.

**4G**). The MDA levels were reduced in the treated groups and more efficacious results were seen with combination treatment (**Fig 4E**) According to our findings, the inflammatory markers showed significant increase in fold change in IL-6 (7.73 ± 0.4), IL-8 (4.24 ± 0.19) and NF-κB (2.94 ± 0.01) gene expression levels in HFFD rats when compared with normal control group, IL-6 (1.02 ± 0.108), IL-8 (1.01 ± 0.07) and NF-κB (1.0 ± 0.03). As shown in **Fig 4H–4J** the treatment groups reduced the expression of inflammatory markers in the liver tissue, and combination therapy decreased gene expression levels in rat livers more significantly (p< 0.001) and the results are comparable to metformin.

**Histopathology evaluation.** Histological examination of the HFFD rat livers revealed focal vacuolar degeneration and raptured blood vessels infiltration of leukocytes and sinusoidal congestion (**Fig 5B**). Histological examination of the metformin treated rat showed mild inflammatory infiltration in portal area (**Fig 5C**). The liver architecture of rutin treated group presented mild hepatocyte vacuolation and most of the morphology appeared normal during evaluation (**Fig 5D**). The vitamin A treated livers of the rats revealed focal hepatic sinusoidal congestion (**Fig 5E**) and the histological examination of rutin, and vitamin A combination exhibited normal morphology (**Fig 5F**). Our results were also supported by histopathology of pancreas of normal control group which revealed normal islet (green arrow in **Fig 5G**). The pancreatic examination of HFFD rat showed shrunken islet and depleted blood vessels whereas the metformin treated rats had pancreas with preserved architecture and islets showed normal morphology. The histological examination of rutin and vitamin A treated pancreas also exhibited normal cellular morphology (green arrow, **Fig 5H** and **5I**). However, the vitamin A treated pancreas showed shrunken islet when compared with NC. The pancreas of the rats receiving combination of rutin, and vitamin A showed normal morphology of the endocrine pancreas (green arrow). The beneficial effect of combination was also supported by our biochemical and PCR results.

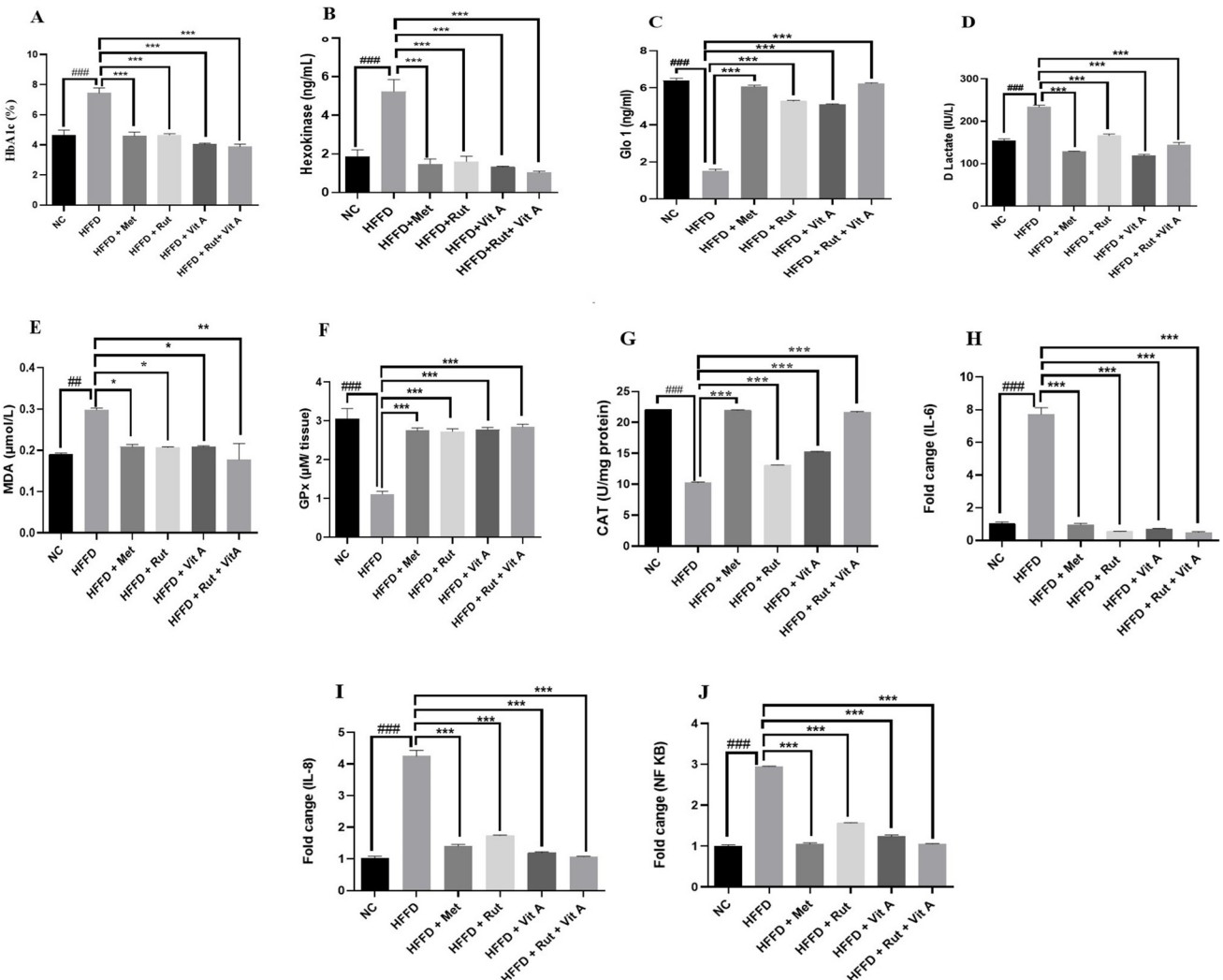

**Fig 4. Effect of Metformin (120mg/kg), Rutin (100mg/kg), Vitamin A (43IU/kg), Rutin (100mg/kg) + Vitamin A (43IU/kg) on AGE biomarkers, oxidative stress markers and gene expression of inflammatory markers in HFFD fed rats.**

## Discussion

It has been established that mental illness, stress, insulin resistance, and diabetes are all linked to hyperglycemia. Consuming sugar-rich diets for an extended period of time as well as fried and canned foods may result in a steady rise in blood sugar levels. [37]. In the current study, we focused on the ability of rutin and vitamin A to improve the non-enzymatic protein glycation, AGE buildup, oxidative stress, and inflammation caused by ongoing hyperglycemia. Rutin's and vitamin A impact on AGE buildup has been studied, however there is little information in the literature about their impact on the hexokinase and glyoxalase system [38]. In the present study, HFFD group showed altered levels of AGE associated parameters like HbA1c, D-lactate, hexokinase 2 and Glo1. Diets heavy in fat and fructose can alter the structure of cells, and fructose has ten times more potential to create AGEs than glucose. Excessive intake of fructose potentiates Maillard reaction which produces harmful altered proteins. [39]. Hexokinase (HK) is an isoenzyme which catalyzes the conversion of glucose to glucose-6-phosphate, serving an essential role in tissue intermediate metabolism [40]. The HFFD

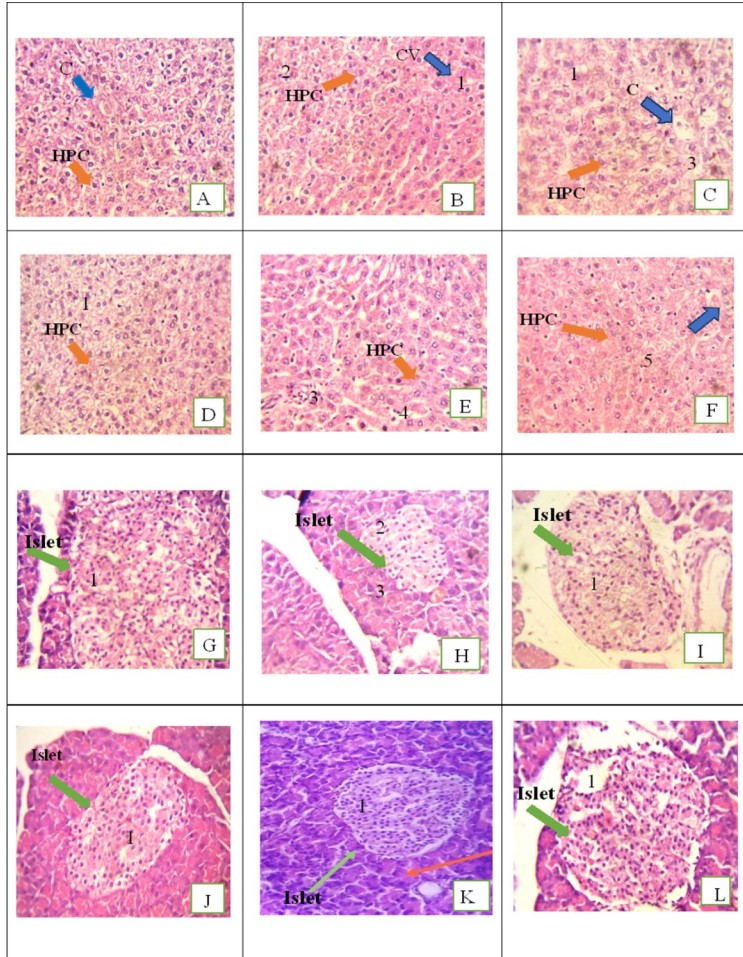

**Fig 5. Photomicrographs of histopathology of liver and pancreas excised from rats provided different treatments.**
A: Normal control, B: HFFD, C: HFFD + Metformin, D: HFFD + Rutin, E: HFFD + Vit A, F: HFFD + Rutin + Vit A.
HPC: Hepatocytes. 1: Focal vacuolation of hepatocytes, 2: Raptured blood vessels, 3: Inflammatory infiltrates, 4:
Hepatic sinusoidal congestion, 5: Normal morphology. G: Normal control, H: HFFD, I: HFFD + Metformin, J: HFFD
+ Rutin, K: HFFD and Vit A, L: HFFD + Rutin + Vit A. 1: Normal islet, 2: Depleted blood vessels, 3: Shrunken islet:
Green arrows present islets.

group demonstrated higher hexokinase levels (**Fig 4B**), and the treatments lowered the hexoki-
nase levels and these findings may be related to those of a prior study in which resveratrol
reversed the elevated hexokinase levels of STZ-induced diabetic rats by protecting the hepatic
and extrahepatic tissues and by increasing insulin release [41]. The *in-silico* study also showed
high binding affinity of rutin with hexokinase (**Table 2**) thereby confirming downregulation
of the enzyme and decrease in AGEs. Glo1 was significantly reduced in HFFD group (**Fig 4C**).
Rutin and vitamin A treatment together resulted in the enzyme's expression being increased,
and the effects are comparable to those of the standard, metformin. Glo1 catalyzes the conver-
sion of hemithioacetal to S-D-lactoylglutathione. Hemithioacetal is formed with the reaction
of methylglyoxal (AGE product) with GPx. These findings imply that elevated Glo1 expression
caused a decrease in MG levels. Additionally, prior research has demonstrated that the highly
expressed and active Glo1 protein prevents the development of AGEs caused by high blood
glucose [42]. The increased expression of Glo1 could have occurred probably by targeting the
Nrf2/ARE pathway [43]. In healthy tissues, the interconversion of pyruvate into lactate, which

occurs in both the mitochondria and cytosol, produces energy under anaerobic conditions. [44]. It is crucial for a significant percentage of the pyruvate and NADH produced by glycolysis to be subsequently oxidized by mitochondria, which is why lactate dehydrogenase activity is reduced in tissues and D-lactate levels are increased. A prior work on resveratrol therapy in diabetic rats provides support for our findings regarding the D-Lactate biomarker (**Fig 4D**), which showed that LDH activity decreased to normal. This was most likely accomplished by regulating the ratio of pyruvate and NADH, which in turn promoted the mitochondrial oxidation of glucose [41,45]. Antioxidant parameters assessed in the current study showed that rutin and vitamin A combination treated rats had reduced MDA levels (**Fig 4E**) and increased GPx and catalase levels as compared to rutin and vitamin A treatments separately. Antioxidant enzymes like GPx and CAT entrap the reactive oxygen species and reduce oxidative stress. An earlier work on rutin suggested that oral administration of rutin enhanced GPx and catalase levels and decreased MDA levels in STZ-induced diabetic mice. Reactive oxygen species are captured by antioxidant enzymes like GPx and CAT, which lessen oxidative stress [46]. Many flavonoids possess anti-inflammatory and antioxidant properties as well as anti-glycating properties as proved in the literature [47]. Rutin's anti-hyperglycemic and free radical scavenging activity is well documented in the literature [46], although its impact on Glo1, hexokinase, and D-lactate as well as anti-inflammatory markers in hyperglycemic state is less well documented. The IL-6, IL-8 and NF-κB are expressed highly during inflammation associated with hyperglycemia. The expression levels of markers were determined in liver tissue homogenate excised from all rats in various groups at the end of study. In the treatment group's significant decrease in fold change in IL-6, IL-8 and NF-κB gene expression levels were observed receiving relevant treatments. The results are comparable with standard metformin and presented in **Fig 4H–4J**. The enhanced genetic expression is also supported by the docking study where rutin exhibited the highest binding affinity with IL-6, with a value of -7.9 kcal/mol, followed by NF-κB, with a value of -7 kcal/mol (**Table 2** and **Fig 1**). The binding affinity of a compound to a protein target refers to the strength of the interaction between them. It is measured in terms of the free energy released upon the formation of the complex, typically expressed in units of kcal/mol. The lower the value of the binding energy, the stronger the binding affinity between the compound and the protein target. The higher binding affinity of rutin towards IL-6 could be due to the presence of specific chemical groups or functional moieties in the compound that are complementary to the active site of IL-6, allowing for a stronger and more stable interaction between them. The higher binding affinity of rutin towards IL-6 compared to NF-κB observed in this study suggests that rutin could be a potential therapeutic agent for IL-6-mediated acute inflammatory diseases. The binding affinity of Vitamin A for different ligands was found to be less than -7 (**Table 2**, **Fig 2**). However, the combination of vitamin A with rutin demonstrated a more efficient ability to reduce the expression of inflammatory cytokines and enhance the levels of antioxidant enzymes.Increase in gene expression level of pro-inflammatory cytokines was evident in literature with HFFD induction and expression of pro-inflammatory cytokines was decreased with flavonoid treatment (e.g. Quercetin and hesperidin) [48]. Rutin exhibits strong antioxidant potential by decreasing the expression of inflammatory cytokines as reported in literature [49]. In the current study rutin reduced the formation of AGEs due to the downregulation of the expression of RAGE and NF-κB and decreased production of IL-6 and IL-8. The *in-vitro* inhibition of fructosamine was performed using BSA-fructose system and the results showed that combination of rutin (100μM) and vitamin A (1200μg/ml) significantly inhibited the fructosamine (amadori product) levels in BSA fructose system as compared to rutin and vitamin A alone. Fructosamine assay is a spectrophotometric approach that relies on ability of Amadori products to convert NBT into monoformazan at 530 nm [50]. The findings of this study can be compared to those of earlier research, which found that the

rhizome of *Alpinia zerumbet* inhibited the glycation of amino groups, prevented the synthesis of Amadori products, and inhibited the synthesis of dicarbonyl compounds, which are produced by the oxidation of proteins or the breakdown of glucose [51]. Our study complied with a previous study proposing that rutin with its free radical scavenging capability is effective in diabetic rats to decrease levels of glycated hemoglobin [52]. This study found that combining vitamin A with rutin treatment resulted in reduction of HbA1c levels (**Fig 3**) thereby suggesting that the combination may be more effective in ameliorating insulin resistance. Our findings were further corroborated by the liver's histology. The liver tissues of rats that had received HFFD treatment exhibited localized hepatocyte degeneration, leukocyte infiltration, and sinusoidal congestion (**Fig 5B**). The treated groups had mild to severe hepatocyte degeneration, chronic inflammatory infiltration in the portal veins, and sinusoidal congestion. (**Fig 5C** and **5D**). Rats given a combination of treatments display a normal liver morphology (**Fig 5E**). According to **Fig 5G**, the histological study of the pancreas from the HFFD group revealed diminished islets and depleted blood vessels in islets. This was corrected by therapy with rutin, vitamin A, and rutin and vitamin A together (**Fig 5J–5L**).

The growing interest in bioactive compounds has been documented in recent years due to their effectiveness, safety, and less cost. Natural bioactive chemicals diminish AGE associated disorders as a result of their anti-glycation properties [39,53] Rutin together with vitamin A may effectively reduce hexokinase levels, ROS, AGE, and RAGE by decreasing IL-6, IL-8, and NF-κB while raising Glo1 expression.

## Conclusion

Rutin and Vitamin A are plant-based bioactive compouds possesing multiple therapeutic benefits. The present study has put forth a compelling proposition that the beneficial effect of rutin and vitamin A could serve as a promising therapeutic approach to combat AGE-associated diseases (diabetes, parkinson's, autism). The *in-silico* findings corroborate with the experimental results, demonstrating the potential of the rutin-vitamin A combination in mitigating glycation-induced damage, dicarbonyl stress, inflammatory cytokine expression, and oxidative stress. Taken together, the results of this study provide valuable insights into the therapeutic potential of the rutin-vitamin A combination for the management of AGE-associated disorders. Further research should be undertaken on rutin and vitamin A to explore their potential for developing therapeutic approaches for diseases associated with advanced glycation end products (AGE).

## Supporting information

**S1 Fig. Oral Glucose Tolerance assessment in Normal Control, HFFD Metformin (120mg/kg), Rutin (100mg/kg), Vitamin A (43IU/kg), Rutin (100mg/kg) + Vitamin A (43IU/kg).** (DOCX)

**S1 Table. *In-silico* studies expressing binding affinity (kcal/mol), hydrogen binding, hydrophobic and electrostatic interactions with distances in Angstrom for investigated ligand Rutin with target proteins.** (DOCX)

**S2 Table. Binding affinity (kcal/mol), hydrogen binding, hydrophobic and electrostatic interactions with distances in Angstrom for investigated ligand Vitamin A with target proteins.** (DOCX)

## Author Contributions

**Conceptualization:** Sairah Hafeez Kamran.

**Data curation:** Aqsa Iqbal.

**Formal analysis:** Aqsa Iqbal.

**Investigation:** Aqsa Iqbal.

**Methodology:** Sairah Hafeez Kamran.

**Project administration:** Sairah Hafeez Kamran.

**Resources:** Saiqa Ishtiaq, Misbah Hameed, Mobina Manzoor.

**Software:** Farhan Siddique.

**Supervision:** Sairah Hafeez Kamran.

**Validation:** Farhan Siddique.

**Writing – original draft:** Aqsa Iqbal.

**Writing – review & editing:** Sairah Hafeez Kamran, Saiqa Ishtiaq, Misbah Hameed, Mobina Manzoor.

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
