## [Decision Letter · Decision Letter 0]

12 Feb 2024

PONE-D-23-43396Modulatory effects of Rutin and Vitamin A on Hyperglycemia induced Glycation, Oxidative stress and Inflammation in High-Fat-Fructose Diet Animal ModelPLOS ONE

Dear Dr. Hafeez Kamran,

Thank you for submitting your manuscript to PLOS ONE. After careful consideration, we feel that it has merit but does not fully meet PLOS ONE’s publication criteria as it currently stands. Therefore, we invite you to submit a revised version of the manuscript that addresses the points raised during the review process.

In this manuscript, authors have studied entitled “Modulatory effects of Rutin and Vitamin A on Hyperglycemia induced Glycation, Oxidative stress and Inflammation in High-Fat-Fructose Diet Animal Model” and have reported that combined effects of rutin and vitamin A in treated rats significantly reduces the level of  HbA1c, hexokinase-2, and D-lactate levels and decreases the level of MDA, CAT, and GPx; and expression of inflammatory cytokines such as IL-6, IL-8, and NF-κB in treated rats. The research findings also indicate that rutin had a strong affinity for several important biomolecules, including as NF-κB, Catalase, MDA, IL-6, hexokinase-2, and GPx which can be hope for the treatment of AGE-related disorders, such as diabetes, autism, Alzheimer’s, atherosclerosis etc.

Overall, the manuscript in its current form is not acceptable for publication in the esteemed “Plos One” journal and requires major revision in the manuscript. Author should revise the manuscript as per reviewer’s comments and can resubmit for the publication in Plos One journal.  

We look forward to receiving your revised manuscript.

Kind regards,

Pankaj Singh, Ph.D.

Academic Editor

PLOS ONE

3. To comply with PLOS ONE submissions requirements, in your Methods section, please provide additional information regarding the experiments involving animals and ensure you have included details on (1) methods of sacrifice, (2) methods of anesthesia and/or analgesia, and (3) efforts to alleviate suffering.

4. Please amend the manuscript submission data (via Edit Submission) to include author Mobina Manzoor.

5. Please amend your authorship list in your manuscript file to include author Mobeena Manzoor.

Reviewers' comments:

Reviewer's Responses to Questions

**Comments to the Author**

1. Is the manuscript technically sound, and do the data support the conclusions?

Reviewer #1: Yes

Reviewer #2: Yes

Reviewer #3: Yes

2. Has the statistical analysis been performed appropriately and rigorously? 

Reviewer #1: Yes

Reviewer #2: Yes

Reviewer #3: Yes

3. Have the authors made all data underlying the findings in their manuscript fully available?

Reviewer #1: Yes

Reviewer #2: Yes

Reviewer #3: Yes

4. Is the manuscript presented in an intelligible fashion and written in standard English?

Reviewer #1: Yes

Reviewer #2: Yes

Reviewer #3: Yes

5. Review Comments to the Author

Reviewer #1: Authors did a good and timely work and represented in very lucid and scientific languages. They represented all the details. Introduction, Materials methods, result and discussion sections are properly represented. However a few mistakes that I observed, that need to be corrected.

1. Line no 59, 60, 62: objectives need to be corrected. sentence formation.

2. Line no: 92: 3 deoxyglucocone should be 3 deoxyglucosone

3,Line 111-120: References are missing

4. Line 214: Slight modification in method: need to be mention.

Please do the correction and submit.

Reviewer #2: This article examines the impact of a combination of rutin and Vitamin D on glycation resulting from hyperglycemia. Primarily, in the present analysis of studies on a mouse model of hyperglycemia involving 30 animals divided into six groups (N=5), endpoints including glycation, oxidative stress, inflammatory response, and cellular integrity were evaluated. The results support that the combination therapy has beneficial effects which are additive and potentially synergistic. An in-silico study showing the comparative binding affinities of the two drugs to identical ligands is also provided and provides further support for the observed experimental outcomes.

This is a well-designed study; the results are presented well and produce a good story. While it the authors may claim that the combination of the drugs has enhanced effects, it may be too early, without further evidence, to claim synergism. The authors are encouraged to be cautious about this claim (synergism).

General comments

Abstract:

Line 71 – 72: Authors must note that NF kappa B is not a cytokine.

Article:

• Lines 197-200 (page 8) should be broken down as the description of the procedure at this point is somewhat ambiguous.

• Line 298 – the figure reference is incomplete.

• Lines 314 – 317 may be broken down as follows:

The pancreas of the rats receiving a combination of rutin and vitamin A showed normal morphology of the endocrine pancreas (green arrow), thereby suggesting a synergistic effect of the combination. This is also supported by our biochemical and PCR results.

Reviewer #3: The manuscript submitted for the publication in “Plos One” Journal is not acceptable in its current form and require major revision. Please refer following comments to improve the manuscript.

Comments:

1. Page 12, Lines 114-118, authors have written that Rutin (quercetin-3-O-rutinoside) is referred as vitamin P, but vitamin P represents Flavonoids or bioflavonoids, a vast group of polyphenolic compounds.

2. In Daily administration of HFFD (30% fat (5% sunflower oil+ 10% cottonseed oil+ 15% saturated fatty acids), why authors have selected combination of different oil instead of pure single oil. What was the basis of their % selection and combination of different oil?

3. Page 15, Lines 187-188, authors should mention that to which groups of animal 10% fructose in drinking water was also given daily for 42 days.

4. Please correct the title of Table 2. Authors mentioned only Binding affinity only for Rutin with target proteins but table has both Rutin and Vitamin A.

5. In methodology section, authors can divide experimental parameters under In vitro, in vivo, and in silico experiments subheads for better understanding of experimental design.

6. In methodology section, In-vitro Fructosamine Assay experimental procedure is incomplete. Authors should also mention the process of Glycation of Bovine Serum Albumin (BSA) weather they have followed the same procedure for Glycation of BSA as in cited reference 23.

7. Page 21, Lines 340, What is MG?

8. Please shift the levelling the Figure 3 (A, B, C……) outside the area of Figure.

9. Page 32, Line 621, Please check the statement ***: p < 0.001 (when compare with HFFD) but HFFD is not showed in Figure.

10. Authors have mentioned in many experimental results that Rutin or Vitamin A or combination of Rutin or Vitamin A showed better or similar protective effects that the synthetic Metformin. Rutin or Vitamin A are well known compounds from long years then why we are not using natural Rutin or Vitamin A for diabetic patients or to cure other diseases instead of synthetic Metformin still Metformin has many side effects.

11. Authors can improve the image quality of Figure 3 as significant level should be more visible and clear.

12. In Molecular docking study, Authors have discussed the results of Rutin more deeply whereas results of Vitamin A are not discussed very effectively.

13. In Figure’s caption and Table’s title, change case should be uniform.

14. Please remove the word discussion from Results and Discussion heading as authors have given separate heading for Discussion section.

15. Authors should strictly follow Instructions for Authors of Plos One’s Journal.

6. PLOS authors have the option to publish the peer review history of their article (what does this mean?). If published, this will include your full peer review and any attached files.

Reviewer #1: **Yes: **SIRSHENDU CHATTERJEE

Reviewer #2: **Yes: **Monde Ntwasa

Reviewer #3: No

---

## [Author Response · Author response to Decision Letter 0]

20 Mar 2024

Dear Editors and Reviewers,

We appreciate you giving us the chance to revise our manuscript, "Modulatory effects of Rutin and Vitamin A on Hyperglycemia induced Glycation, Oxidative stress, and Inflammation in High-Fat-High-Fructose Diet Animal Model." Your insightful and beneficial comments have helped us to improve our manuscript. Every comment has been thoroughly reviewed, and the appropriate modifications have been made. We trust that all the modifications have been carried out correctly, and we hope to receive approval. All the corrections and responses to reviewer comments are as follows.

Response to Journal Requirements

Response: Thank you for your valuable suggestion. We have carefully revised the manuscript according to the instructions provided.

Response: We have applied codes that follow the best practice and facilitates reproducibility and reuse.

3. To comply with PLOS ONE submissions requirements, in your Methods section, please provide additional information regarding the experiments involving animals and ensure you have included details on (1) methods of sacrifice, (2) methods of anesthesia and/or analgesia, and (3) efforts to alleviate suffering.

Response: Thank you for your valuable suggestion. We have included the method of sacrifice and method of anaesthesia in the study design section.

Response: Thank you for your valuable suggestion. We have included all data in the manuscript and remaining data in the Supplementary file attached.

5. Please amend the manuscript submission data (via Edit Submission) to include author Mobina Manzoor.

Response: Thank you for your valuable suggestion. We have amended the name Mobina Manzoor. 

6. Please amend your authorship list in your manuscript file to include author Mobeena Manzoor.

Response: Thank you for your valuable suggestion. We have amended the name Mobeena Manzoor to Mobina Manzoor in the authorship list.

7. Your ethics statement should only appear in the Methods section of your manuscript. If your ethics statement is written in any section besides the Methods, please move it to the Methods section and delete it from any other section. Please ensure that your ethics statement is included in your manuscript, as the ethics statement entered into the online submission form will not be published alongside your manuscript.

Response: Thank you for your valuable suggestion. We have deleted the Ethical statement from the declaration section and moved to the methods section.

Response to Reviewers

Reviewer #1

Authors did a good and timely work and represented in very lucid and scientific languages. They represented all the details. Introduction, Materials methods, result and discussion sections are properly represented. However a few mistakes that I observed, that need to be corrected.

Response: We are grateful for your encouraging remarks. The responses of the questions are provided.

Line no 59, 60, 62: objectives need to be corrected. sentence formation.

Response: We have reformed the sentence in Line 59-62

In the current study we investigated the impact of combination of rutin and vitamin A on glycated products, the glyoxalase system, oxidative markers, and inflammation in animals fed a high-fat high-fructose (HFFD) diet.

Line no: 92: 3 deoxyglucocone should be 3 deoxyglucosone

Response: Thank you for your insightful suggestion. We have rectified to 3-deoxyglucosone.

Line 111-120: References are missing

Response: Thank you for your valuable suggestion. We have added reference in line 117.

Line 214: Slight modification in method: need to be mention.

Response: Thank you for helpful comment. We have added the methods.

Reviewer #2

This article examines the impact of a combination of rutin and Vitamin D on glycation resulting from hyperglycemia. Primarily, in the present analysis of studies on a mouse model of hyperglycemia involving 30 animals divided into six groups (N=5), endpoints including glycation, oxidative stress, inflammatory response, and cellular integrity were evaluated. The results support that the combination therapy has beneficial effects which are additive and potentially synergistic. An in-silico study showing the comparative binding affinities of the two drugs to identical ligands is also provided and provides further support for the observed experimental outcomes.

Response: We appreciate your supportive remarks.

This is a well-designed study; the results are presented well and produce a good story. While it the authors may claim that the combination of the drugs has enhanced effects, it may be too early, without further evidence, to claim synergism. The authors are encouraged to be cautious about this claim (synergism).

Response: Thank you for your valuable suggestions. We have replaced the word “synergism” with “beneficial’.

Abstract:

Line 71 – 72: Authors must note that NF kappa B is not a cytokine.

Response: Thank you for your valuable suggestion. We have corrected the statement.

Article

• Lines 197-200 (page 8) should be broken down as the description of the procedure at this point is somewhat ambiguous.

• Line 298 – the figure reference is incomplete.

• Lines 314 – 317 may be broken down as follows:

The pancreas of the rats receiving a combination of rutin and vitamin A showed normal morphology of the endocrine pancreas (green arrow), thereby suggesting a synergistic effect of the combination. This is also supported by our biochemical and PCR results.

Response: We have broken down the statements to decrease the ambiguity.

Line 298: We have corrected the figure reference.

Line 314-317: Thank you for your valuable suggestion. We have broken down the statement in Line 314-317.

Reviewer #3: 

The manuscript submitted for the publication in “Plos One” Journal is not acceptable in its current form and require major revision. Please refer following comments to improve the manuscript.

Response: We have made every effort to amend the content in accordance with the helpful recommendations given.

Page 12, Lines 114-118, authors have written that Rutin (quercetin-3-O-rutinoside) is referred as vitamin P, but vitamin P represents Flavonoids or bioflavonoids, a vast group of polyphenolic compounds.

Response: Thank you for your valuable suggestion. We have corrected the statement. We found rutin referred to as vitamin P in an article (https://www.ncbi.nlm.nih.gov/pmc/articles/PMC5355559/. But Vitamin P broadly represents more than one flavonoid. We have removed the word Vitamin P from page 12.

In Daily administration of HFFD (30% fat (5% sunflower oil+ 10% cottonseed oil+ 15% saturated fatty acids), why authors have selected combination of different oil instead of pure single oil. What was the basis of their % selection and combination of different oil?

Response: The HFFD utilized in this research was specifically formulated to induce hyperglycemia, and the cited reference was studied. Various oils were utilized to create a blend of unsaturated and saturated fatty acids. Sunflower and cotton seed oils both include a combination of unsaturated and saturated fatty acids. 15% saturated fatty acids were introduced separately to induce metabolic imbalance and obesity.

Page 15, Lines 187-188, authors should mention that to which groups of animal 10% fructose in drinking water was also given daily for 42 days.

Response: Thank you for your valuable suggestion. We have added the statement in the study design.

Please correct the title of Table 2. Authors mentioned only Binding affinity only for Rutin with target proteins but table has both Rutin and Vitamin A.

Response: Thank you for your valuable suggestion. We have corrected the title of the table 2.

In methodology section, authors can divide experimental parameters under In vitro, in vivo, and in silico experiments subheads for better understanding of experimental design.

Response: We have divided the sections as mentioned in the comment.

In methodology section, In-vitro Fructosamine Assay experimental procedure is incomplete. Authors should also mention the process of Glycation of Bovine Serum Albumin (BSA) weather they have followed the same procedure for Glycation of BSA as in cited reference 23.

Response: Thank you for your valuable comment. We have added the method of glycation of bovine serum albumin in the in-vitro fructosamine assay.

Page 21, Lines 340, What is MG?

Response: We have added the abbreviation of methylglyoxal (MG) in the abbreviation list and full name along with abbreviation is mentioned in Line 95 in Introduction

Please shift the levelling the Figure 3 (A, B, C……) outside the area of Figure.

Response: Thank you for the helpful comment. We have shifted the levelling of the figure 3 outside the figure.

Page 32, Line 621, Please check the statement ***: p < 0.001 (when compare with HFFD) but HFFD is not showed in Figure.

Response: Thank you for your helpful comment. We have corrected the statement below the Fig. 2.

Authors have mentioned in many experimental results that Rutin or Vitamin A or combination of Rutin or Vitamin A showed better or similar protective effects that the synthetic Metformin. Rutin or Vitamin A are well known compounds from long years then why we are not using natural Rutin or Vitamin A for diabetic patients or to cure other diseases instead of synthetic Metformin still Metformin has many side effects.

Response: Thank you for your valuable comment. 

In the current investigation, the combination of rutin and vitamin A demonstrated benefits that were comparable to those of metformin. The evidence for substituting rutin and vitamin A combination for metformin is insufficient. 

Although the purpose of this study was to support the previously published findings, we are still unable to assert that a combination of rutin and vitamin A may replace metformin. 

We made the necessary corrections to the conclusion section and suggested that more research be done on the combination of rutin and vitamin A for the possible development of therapeutic approaches for conditions associated with advanced glycation end products (AGE).

Authors can improve the image quality of Figure 3 as significant level should be more visible and clear.

Response: Thank you for your valuable suggestion. We have improved the quality of images.

In Molecular docking study, Authors have discussed the results of Rutin more deeply whereas results of Vitamin A are not discussed very effectively.

Response: We have added the results of vitamin A in the results and discussion section

In Figure’s caption and Table’s title, change case should be uniform.

Response: We are grateful for your helpful suggestion. The change case has been uniformed in the titles of figures and tables.

Please remove the word discussion from Results and Discussion heading as authors have given separate heading for Discussion section.

Response: Thank you. We have removed the word from the heading.

Authors should strictly follow Instructions for Authors of Plos One’s Journal.

Response: Thank you for your valuable suggestion. We have tried to strictly follow the instructions for authors of Plos One.

---

## [Editor Report · Decision Letter 1]

27 Mar 2024

PONE-D-23-43396R1Modulatory effects of Rutin and Vitamin A on hyperglycemia induced glycation, oxidative stress and inflammation in high-fat-fructose diet animal modelPLOS ONE

Dear Dr. Hafeez Kamran,

Thank you for submitting your manuscript to PLOS ONE. After careful consideration, we feel that it has merit but does not fully meet PLOS ONE’s publication criteria as it currently stands. Therefore, we invite you to submit a revised version of the manuscript that addresses the points raised during the review process.

The revised manuscript can be accepted for publication in esteemed "Plos One” Journal with few minor corrections. Authors have addressed each queries very well raised by reviewer and have critically modify the manuscript as per requirement. Following corrections still need to improve the submitted manuscript.

Comments:

1. Table 3, Unit should be uniform throughout Table.

2. Please improve the quality of Fig 1, 2 and 5 as words are not visible to read. It is requested that authors should submit the revised manuscript with improved figures.

3.  Please correct the order of Figures in submitted manuscript.

We look forward to receiving your revised manuscript.

Kind regards,

Pankaj Singh, Ph.D.

Academic Editor

PLOS ONE
---

## [Author Response · Author response to Decision Letter 1]

16 Apr 2024

Dear Editors and Reviewers,

We appreciate you giving us the chance to revise our manuscript, "Modulatory effects of Rutin and Vitamin A on Hyperglycemia induced Glycation, Oxidative stress, and Inflammation in High-Fat-High-Fructose Diet Animal Model." Your insightful and beneficial comments have helped us to improve our manuscript. Every comment has been thoroughly reviewed, and the appropriate modifications have been made. We trust that all the modifications have been carried out correctly, and we hope to receive approval. All the corrections and responses to reviewer comments are as follows.

Response to Academic Editor Comments

Table 3, Unit should be uniform throughout Table.

Response: The Table has been corrected.

Please improve the quality of Fig 1, 2 and 5 as words are not visible to read. It is requested that authors should submit the revised manuscript with improved figures.

Response: The quality of the figures 1, 2 and 5 have been improved. We have uploaded the figures on Preflight Analysis and Conversion Engine (PACE) digital diagnostic tool and ensured that figures meet Plos requirements.

Please correct the order of Figures in submitted manuscript.

The order of the figures has been corrected.

Response to Journal Requirements

Response: Thank you for your valuable suggestion. We have carefully revised the manuscript according to the instructions provided.

Response: We have applied codes that follow the best practice and facilitates reproducibility and reuse.

3. To comply with PLOS ONE submissions requirements, in your Methods section, please provide additional information regarding the experiments involving animals and ensure you have included details on (1) methods of sacrifice, (2) methods of anesthesia and/or analgesia, and (3) efforts to alleviate suffering.

Response: Thank you for your valuable suggestion. We have included the method of sacrifice and method of anaesthesia in the study design section.

Response: Thank you for your valuable suggestion. We have included all data in the manuscript and remaining data in the Supplementary file attached.

5. Please amend the manuscript submission data (via Edit Submission) to include author Mobina Manzoor.

Response: Thank you for your valuable suggestion. We have amended the name Mobina Manzoor. 

6. Please amend your authorship list in your manuscript file to include author Mobeena Manzoor.

Response: Thank you for your valuable suggestion. We have amended the name Mobeena Manzoor to Mobina Manzoor in the authorship list.

7. Your ethics statement should only appear in the Methods section of your manuscript. If your ethics statement is written in any section besides the Methods, please move it to the Methods section and delete it from any other section. Please ensure that your ethics statement is included in your manuscript, as the ethics statement entered into the online submission form will not be published alongside your manuscript.

Response: Thank you for your valuable suggestion. We have deleted the Ethical statement from the declaration section and moved to the methods section.

Response to Reviewers

Reviewer #1

Authors did a good and timely work and represented in very lucid and scientific languages. They represented all the details. Introduction, Materials methods, result and discussion sections are properly represented. However a few mistakes that I observed, that need to be corrected.

Response: We are grateful for your encouraging remarks. The responses of the questions are provided.

Line no 59, 60, 62: objectives need to be corrected. sentence formation.

Response: We have reformed the sentence in Line 59-62

In the current study we investigated the impact of combination of rutin and vitamin A on glycated products, the glyoxalase system, oxidative markers, and inflammation in animals fed a high-fat high-fructose (HFFD) diet.

Line no: 92: 3 deoxyglucocone should be 3 deoxyglucosone

Response: Thank you for your insightful suggestion. We have rectified to 3-deoxyglucosone.

Line 111-120: References are missing

Response: Thank you for your valuable suggestion. We have added reference in line 117.

Line 214: Slight modification in method: need to be mention.

Response: Thank you for helpful comment. We have added the methods.

Reviewer #2

This article examines the impact of a combination of rutin and Vitamin D on glycation resulting from hyperglycemia. Primarily, in the present analysis of studies on a mouse model of hyperglycemia involving 30 animals divided into six groups (N=5), endpoints including glycation, oxidative stress, inflammatory response, and cellular integrity were evaluated. The results support that the combination therapy has beneficial effects which are additive and potentially synergistic. An in-silico study showing the comparative binding affinities of the two drugs to identical ligands is also provided and provides further support for the observed experimental outcomes.

Response: We appreciate your supportive remarks.

This is a well-designed study; the results are presented well and produce a good story. While it the authors may claim that the combination of the drugs has enhanced effects, it may be too early, without further evidence, to claim synergism. The authors are encouraged to be cautious about this claim (synergism).

Response: Thank you for your valuable suggestions. We have replaced the word “synergism” with “beneficial’.

Abstract:

Line 71 – 72: Authors must note that NF kappa B is not a cytokine.

Response: Thank you for your valuable suggestion. We have corrected the statement.

Article

• Lines 197-200 (page 8) should be broken down as the description of the procedure at this point is somewhat ambiguous.

• Line 298 – the figure reference is incomplete.

• Lines 314 – 317 may be broken down as follows:

The pancreas of the rats receiving a combination of rutin and vitamin A showed normal morphology of the endocrine pancreas (green arrow), thereby suggesting a synergistic effect of the combination. This is also supported by our biochemical and PCR results.

Response: We have broken down the statements to decrease the ambiguity.

Line 298: We have corrected the figure reference.

Line 314-317: Thank you for your valuable suggestion. We have broken down the statement in Line 314-317.

Reviewer #3: 

The manuscript submitted for the publication in “Plos One” Journal is not acceptable in its current form and require major revision. Please refer following comments to improve the manuscript.

Response: We have made every effort to amend the content in accordance with the helpful recommendations given.

Page 12, Lines 114-118, authors have written that Rutin (quercetin-3-O-rutinoside) is referred as vitamin P, but vitamin P represents Flavonoids or bioflavonoids, a vast group of polyphenolic compounds.

Response: Thank you for your valuable suggestion. We have corrected the statement. We found rutin referred to as vitamin P in an article (https://www.ncbi.nlm.nih.gov/pmc/articles/PMC5355559/. But Vitamin P broadly represents more than one flavonoid. We have removed the word Vitamin P from page 12.

In Daily administration of HFFD (30% fat (5% sunflower oil+ 10% cottonseed oil+ 15% saturated fatty acids), why authors have selected combination of different oil instead of pure single oil. What was the basis of their % selection and combination of different oil?

Response: The HFFD utilized in this research was specifically formulated to induce hyperglycemia, and the cited reference was studied. Various oils were utilized to create a blend of unsaturated and saturated fatty acids. Sunflower and cotton seed oils both include a combination of unsaturated and saturated fatty acids. 15% saturated fatty acids were introduced separately to induce metabolic imbalance and obesity.

Page 15, Lines 187-188, authors should mention that to which groups of animal 10% fructose in drinking water was also given daily for 42 days.

Response: Thank you for your valuable suggestion. We have added the statement in the study design.

Please correct the title of Table 2. Authors mentioned only Binding affinity only for Rutin with target proteins but table has both Rutin and Vitamin A.

Response: Thank you for your valuable suggestion. We have corrected the title of the table 2.

In methodology section, authors can divide experimental parameters under In vitro, in vivo, and in silico experiments subheads for better understanding of experimental design.

Response: We have divided the sections as mentioned in the comment.

In methodology section, In-vitro Fructosamine Assay experimental procedure is incomplete. Authors should also mention the process of Glycation of Bovine Serum Albumin (BSA) weather they have followed the same procedure for Glycation of BSA as in cited reference 23.

Response: Thank you for your valuable comment. We have added the method of glycation of bovine serum albumin in the in-vitro fructosamine assay.

Page 21, Lines 340, What is MG?

Response: We have added the abbreviation of methylglyoxal (MG) in the abbreviation list and full name along with abbreviation is mentioned in Line 95 in Introduction

Please shift the levelling the Figure 3 (A, B, C……) outside the area of Figure.

Response: Thank you for the helpful comment. We have shifted the levelling of the figure 3 outside the figure.

Page 32, Line 621, Please check the statement ***: p < 0.001 (when compare with HFFD) but HFFD is not showed in Figure.

Response: Thank you for your helpful comment. We have corrected the statement below the Fig. 2.

Authors have mentioned in many experimental results that Rutin or Vitamin A or combination of Rutin or Vitamin A showed better or similar protective effects that the synthetic Metformin. Rutin or Vitamin A are well known compounds from long years then why we are not using natural Rutin or Vitamin A for diabetic patients or to cure other diseases instead of synthetic Metformin still Metformin has many side effects.

Response: Thank you for your valuable comment. 

In the current investigation, the combination of rutin and vitamin A demonstrated benefits that were comparable to those of metformin. The evidence for substituting rutin and vitamin A combination for metformin is insufficient. 

Although the purpose of this study was to support the previously published findings, we are still unable to assert that a combination of rutin and vitamin A may replace metformin. 

We made the necessary corrections to the conclusion section and suggested that more research be done on the combination of rutin and vitamin A for the possible development of therapeutic approaches for conditions associated with advanced glycation end products (AGE).

Authors can improve the image quality of Figure 3 as significant level should be more visible and clear.

Response: Thank you for your valuable suggestion. We have improved the quality of images.

In Molecular docking study, Authors have discussed the results of Rutin more deeply whereas results of Vitamin A are not discussed very effectively.

Response: We have added the results and discussion of vitamin A in the results and discussion section

In Figure’s caption and Table’s title, change case should be uniform.

Response: We are grateful for your helpful suggestion. The change case has been uniformed in the titles of figures and tables.

Please remove the word discussion from Results and Discussion heading as authors have given separate heading for Discussion section.

Response: Thank you. We have removed the word from the heading.

Authors should strictly follow Instructions for Authors of Plos One’s Journal.

Response: Thank you for your valuable suggestion. We have tried to strictly follow the instructions for authors of Plos One.

---

## [Editor Report · Decision Letter 2]

19 Apr 2024

Modulatory effects of Rutin and Vitamin A on hyperglycemia induced glycation, oxidative stress and inflammation in high-fat-fructose diet animal model

PONE-D-23-43396R2

Dear Dr. Hafeez Kamran,

We’re pleased to inform you that your manuscript has been judged scientifically suitable for publication and will be formally accepted for publication once it meets all outstanding technical requirements.

Kind regards,

Pankaj Singh, Ph.D.

Academic Editor

PLOS ONE
---

## [Editor Report · Acceptance letter]

29 Apr 2024

PONE-D-23-43396R2 

PLOS ONE

Dear Dr. Hafeez Kamran, 

I'm pleased to inform you that your manuscript has been deemed suitable for publication in PLOS ONE. Congratulations! Your manuscript is now being handed over to our production team.

Kind regards, 

on behalf of

Dr. Pankaj Singh 

Academic Editor

PLOS ONE